# Modifying Puree Meals in Residential Aged Care Facilities: A Multi-Centre Feasibility and Acceptability Study

**DOI:** 10.3390/geriatrics6040108

**Published:** 2021-11-05

**Authors:** Dai Pu, Yuen-Yu Choi, Karen Man-Kei Chan, May Man-Wai Poon

**Affiliations:** 1Faculty of Medicine, Nursing and Health Sciences, School of Primary and Allied Health Care, Monash University, Frankston 3199, Australia; 2Swallowing Research Laboratory, Faculty of Education, The University of Hong Kong, Pokfulam, Hong Kong, China; yukichoist@gmail.com (Y.-Y.C.); karencmk@hku.hk (K.M.-K.C.); 3ENT Laser Hearing & Speech Therapy Centre, 59–65 Queen’s Road Central, Hong Kong, China

**Keywords:** deglutition, mealtime difficulties, residential aged care facilities, long term care facilities, nutrition, texture modification, swallowing function, puree

## Abstract

Purees are often recommended for older adults in residential aged care facilities (RACFs) to target swallowing difficulties and nutrition. However, they lack appeal and may have negative impacts on nutritional intake. This study investigated the subjective experiences and objective swallowing function and safety of older adults in response to a modified puree. Twenty-eight residents from three RACFs whose regular diets consisted of purees were recruited. Purees were modified to improve visual appeal by adding a commercial enzyme gellant. Each participant was observed during three puree and three modified puree meals, and completed a brief questionnaire after each meal. A videofluoroscopic swallowing study (VFSS) was performed with 16 of the participants. Compared to purees, modified purees were observed to be easier for oral processing and intake amount was higher, but participants required assistance more often and mealtimes were longer. Participants did not show preference for either type of puree. VFSS showed similar swallowing responses between the two puree types; however, a distinction was observed for modified pureed meat compared to other ingredients. Modifying puree meals in RACFs is a feasible approach to improve nutritional intake while maintaining swallowing safety, but their appeal to consumers is not definitive.

## 1. Introduction

Texture-modified diets (TMDs) consist of foods and drinks that are processed to achieve textures that are easier to eat and swallow for individuals who may experience mealtime difficulties, dentition issues, or swallowing difficulties (dysphagia). TMDs can consist of foods that are naturally easy to swallow, but often require additional preparation to achieve the desired texture. Common methods of food texture modification include softening, mincing, and pureeing. The International Dysphagia Diet Standardization Initiative (IDDSI) [1] addresses the different categories for foods and liquids used for dysphagia management. The purpose of the TMD is to maintain oral nutrition while ensuring maximum swallowing safety, so that individuals do not choke or aspirate while consuming food and drink. In the short-term, they can be a part of the rehabilitation effort until the individual recovers from illness or trauma. In the long-term, such as for older adults who live in residential aged care facilities (RACFs), TMDs can have potentially negative impacts on those they are designed to help.

There are three major negative aspects to the implementation of TMDs. First, although higher viscosity is associated with reduced penetration-aspiration [2], swallowing safety is not guaranteed [3,4,5] as aspiration of food material into the airway can still occur; and post-swallow residue in the pharynx can also increase with viscosity [2]. Second, the nutritional value of TMDs can be compromised. Textures such as purees tend to be less nutritiously dense compared to regular-textured diets [6,7], which may place those who rely on them for sustenance at a higher risk for malnutrition [8]. Lastly, those who eat and prepare TMDs have reported dissatisfaction with all aspects of the foods [8,9,10]. This dissatisfaction may explain the overall reduced nutritional intake of [7], and low adherence to TMDs [11]. Empirical evidence has shown that hospitalised older adults who consume TMDs consume less energy and protein compared to those on regular diets [12]. Those who live in nursing homes also consume less energy and fluids when placed on TMDs [11]. As a result, TMDs are associated with undernutrition overall [13] and reduced quality of life [14].

In response to these shortcomings, attempts have been made to optimise the texture and appeal of TMDs, including optimising visual appearance [15,16,17,18] and flavours [19], using novel texture-modifying agents such as xantham gum to thicken fluids [20], and enzymes [16] and gelling agents [21] to modify solid foods. Initial evidence has shown that these attempts can be successful in increasing safety variables [20,21] and oral intake [7,18], however, the body of evidence remains small. This study aimed to contribute to the research of TMD optimisation by investigating the effects of using a gelling agent in puree foods on swallowing safety and consumers’ acceptance. There were three primary objectives:

### Study Objectives

(1)Investigate the feasibility of administering puree foods that have been modified with a gelling agent in RACFs.(2)Compare RACF residents’ mealtime experiences of traditional puree and modified puree.(3)Compare swallowing safety and efficiency between traditional puree and modified puree when consumed by RACF residents.

## 2. Materials and Methods

This study had two parts. The first part used a multi-centre repeated measures design. Data was collected from participants across three consecutive weeks for traditional puree meals, followed by three consecutive weeks for modified puree meals. The second part of the study was a cross-sectional observational trial that measured the swallowing function of all participants while eating the two different meals using videofluoroscopic swallowing studies (VFSS). This study was chosen to be conducted in Hong Kong, where aspiration can be observed in 35.3% of older adults in the community and the aged care sector [22], and up to 71.1% and 67.2% of those who received aged care services require liquid and food-texture modification, respectively [23]. The Hong Kong government’s recent budget injections into the aged care sector have stimulated interest in dysphagia products and increased their value, but there is no scientific evidence that applies to the local population to guide consumer choice.

### 2.1. Participants

Three RACFs were recruited for this study. Two weeks before data collection, the second author Y.-Y.C., a trained speech therapist, screened residents at each facility for study inclusion. Residents met the inclusion criteria for the study if they showed signs and symptoms of dysphagia and had been eating texture-modified diets that had at least one puree texture (of rice, vegetables, or meat) for at least 6 months prior to the start of the study. Residents who required tube-feeding and were completely dependent for feeding were not considered for the study. Consent was obtained from the participants themselves, or their legal guardian if they were unable to give consent. The study was approved by the Human Research Ethics Committee of The University of Hong Kong (Reference Number: EA2005036).

### 2.2. Modified Puree Meal Preparations

Lunch and dinner meals served in RACFs are usually composed of four components—soup, rice, meat, and vegetables; the latter four are usually served in individual bowls/platters or in the same platter with divided sections for each component. For mealtime observations in this study, the ingredients were rice, pork, cabbage, carrot, and lotus root; for VFSS, the ingredients were rice, pork, and cabbage. Purees were modified using an enzyme gellant (FoodCare Co., Ltd., Sagamihara, Japan) added as a percentage of the net weight of the puree; for minced or soft textures, additional water was added first, followed by the enzyme gellant. To ensure the validity and reliability of the modified puree meals, three steps preceded data collection procedures:(1)Consultation workshop for testing different ratios of ingredients: the final author, M.P., a speech therapist with 20 years of clinical experience in dysphagia management, audited the modified puree textures to ensure they would be consistent across all facilities.(2)Modified puree preparation workshop: a 3 h training workshop was held for the kitchen staff at all RACFs on how to prepare the modified purees according to the standards set in step 1.(3)Pre-serving meal inspection: all meals prepared for this study were required to be inspected and approved by Y.-Y.C. who participated in the first two steps before they were served for data collection.

Traditional purees were assessed against the IDDSI framework for level 4 puree (Figure 1); modified purees were audited to match the texture agreed upon in step 1 to assess feasibility of the RACF kitchen to produce modified purees following training.

The final mixture for modified purees was: 1.5% enzyme gellant in rice, 1.2% enzyme gellant in meat, 1.2% enzyme gellant in cabbage, 1.2% enzyme gellant in carrots, and 1% in lotus root. To modify solid foods that were not in puree form (minced or soft), additional water was added as a percentage of the net weight: 200% in rice, 150% in meat, 70% in cabbage, 70% in carrots, and 100% in lotus root. The combined food, enzyme gellant and/or water were blended and heated in a thermal blender (Robot Coupe Robot Cook, model 43001R) for 4–6 min until the mixture reached 100 °C. The mixture was then poured into silicon molds that mimicked the shapes of the original food ingredients (Figure 2). The mixture took 2 min to solidify and was then served as usual.

The modified puree was categorised as transitional foods using the IDDSI framework. It had a firm texture but would transition into a smooth pureed form after temperature and minimal pressure were applied.

## 3. Procedures

### 3.1. Study Part 1

#### 3.1.1. Procedures

Participants were observed for one of their regular meals per week for three consecutive weeks. Following this, participants were served a modified puree meal for one meal a week for three consecutive weeks and observed. Y.-Y.C. and a speech therapist research assistant observed and rated the first 30 meals together to standardise their ratings. All subsequent meals were observed with one speech therapist observing and rating each participant each meal. Each speech therapist tallied the outcome measures that indicated swallowing or oral intake difficulties; this was accompanied by a brief questionnaire to probe for subjective experience of the meals. The questionnaire was used to measure acceptability of the modified puree meals.

#### 3.1.2. Outcome Measures

For each participant during each meal, the following events were documented:Difficulty picking up food with a spoon;Difficulty bringing food to the lips with a spoon;Difficulty retaining food in the mouth;Food was observed to spill from the lips unintentionally;Oral residue was observed at the end of the meal;Aid was needed from a personal care worker (PCW);Number of times the participant coughed;Number of times the participant cleared their throat;Time taken to complete the meal (minutes), either by finishing the food or indicating the wish to stop;Percentage of meal eaten estimated by the observing speech therapist (the meals were not weighed as food was often spilt onto the table or the floor).

For each meal, the questionnaire asked each participant to first identify the food on their plate and their accuracy was documented. After the meal, they were asked to give their satisfaction on a Likert scale of 1 (very dissatisfied) to 4 (very satisfied) for the following aspects of the meal:Visual appeal;Texture;Taste;Ease of swallowing.

Researchers were unable to elicit consistent responses on the Likert scale for the “ease of swallowing” of meals, so this was amended to a binary yes/no response from participants. The reasons and implications for this are addressed in the Discussion section of this paper.

### 3.2. Study Part 2

#### 3.2.1. Procedures

Participants underwent a single session of videofluoroscopic swallowing study (VFSS) while sitting upright. The barium consisted of 340 g E-Z-HD (98% *w*/*w*) barium and 65 mL of water. Traditional puree trials were prepared by combining liquid barium with traditional puree. Nestle ThickenUp starch-based thickener was then added to obtain an IDDSI level 4 texture and was checked by Y.-Y.C. with a spoon-tilt test. To prepare the modified puree trials, each spoonful was mixed with liquid barium. This was then cut into four smaller pieces to ensure the bolus was sufficiently coated. Two swallows of each component of each meal type were planned, i.e., rice, cabbage, and pork. The order of bolus presentation was planned to be six trials of one meal type (either traditional puree or modified puree), followed by six trials of the other meal type. Fatigue was observed in the first two participants who underwent VFSS; therefore the order of bolus trials was modified to two trials of modified puree rice, two trials of traditional puree rice, two trials of modified puree cabbage, two trials of traditional puree cabbage, two trials of modified puree pork, and two trials of traditional puree pork.

Each participant was spoon-fed; each bolus was 10 mL. If the participant showed any clinical signs of aspiration on two consecutive trials, they were instructed to clear the bolus and progress to the next meal type. If there was any occurrence of silent aspiration and no effective cough was initiated, the session was ceased. If the participant was unable to swallow spontaneously or if the bolus was not cleared after spontaneous swallows, verbal prompts were given by Y.-Y.C. after 5 s.

#### 3.2.2. Outcome Measures

VFSS footage for each swallow was rated by Y.-Y.C. for:Penetration Aspiration Scale (PAS) score [24];Eisenhuber scale of pharyngeal residue [25] score;Number of swallows needed to clear each bolus.

Ten percent of the swallows were independently rated by a second speech therapist to test for inter-rater reliability.

### 3.3. Data Analysis

Mealtime observation difficulties were compiled as a score out of 3 to indicate the number of meals during which they were observed for each meal type. The mean for number of coughs per meal, number of throat clears per meal, time taken to complete meal, and percentage eaten per meal were calculated for each meal type. For participant feedback, the mean satisfaction score was calculated for each meal type, and the mean accuracy of identifying components of each meal type was calculated. For VFSS data, the worst PAS score was used for each swallow, and the mean was calculated for each meal component for each meal type. The mean Eisenhuber residue score and number of swallows were calculated for each meal component for each meal type. Paired samples tests were used to compare each outcome between traditional puree and modified puree; paired samples Wilcoxon tests were used for variables that were not normally distributed. Effect sizes were calculated for variable pairs with statistically significant differences. A *p*-value of less than 0.05 was considered statistically significant.

## 4. Results

A total of 30 RACF residents were recruited from three RACFs, with 10 from each RACF. Two participants passed away following recruitment; the final dataset used for analysis were collected from 28 participants. Seventeen of these participants were female (60.7%), and 11 (39.3%) were male. Their mean age was 86.1 years old (standard deviation = 8.4, range = 72–104). All participants were rated as severely frail on the Clinical Frailty Scale [26], and levels 4 or 5 on the Functional Oral Intake Scale [27]. They required high levels of care and presented with a range of diagnoses in their medical background (Table 1).

### 4.1. Study Part 1

#### 4.1.1. Mealtime Observations

All modified puree meals were prepared under the supervision of trainers who provided the 3 h workshop to kitchen staff. After preparation, these meals were inspected and approved by Y.-Y.C. before serving. No meals required further adjusting.

All 28 participants were observed for both meal types. One participant refused meals on all observation days, three participants were absent for one traditional puree meal each, two participants were absent for one modified puree meal each, and one participant refused one modified puree meal. A total of 156 meals were observed during the 6-week data collection period. 

Table 2 shows the findings of each outcome measure and the statistical comparisons between the meal types for each measure; Wilcoxon tests were used as most outcome measures were not normally distributed, with the exception of percentage of intake and meal time, which were analysed using paired samples t-tests. During mealtime observations, participants eating modified purees had more difficulty scooping up the modified purees with a spoon (*p* = 0.013). There was also less unintentional anterior spillage from the lips (*p* = 0.01). Observed coughing and throat clearing during the two meal types were not significantly different. More of the modified purees were estimated to be eaten by the end of the meal (*p* = 0.05), but each meal also took longer for participants to finish (*p* = 0.014).

#### 4.1.2. Participant Feedback

There were no statistically significant differences between the meals for any of the questionnaire items designed to assess acceptability of the meals by the participants. In general, the participants in this study had difficulty providing responses based on a Likert scale for satisfaction, and often defaulted to binary yes/no responses. As a result, participant satisfaction data could not capture feedback from the entire cohort. Table 3 shows the participant feedback data collected for each meal type.

### 4.2. Study Part 2

VFSS was carried out with 16 participants (57% of recruited cohort) within 3 months of the meal observations in part 1 of the study. The remainder of the cohort were unable to attend VFSS in hospital due to COVID19 restrictions while the study was being conducted. A total of 179 swallows were observed in VFSS.

Table 4 shows the PAS score, residue score, and number of swallows to clear bolus for each component (vegetable, meat, rice) of each meal type. Wilcoxon tests were used as PAS ratings were not normally distributed. Modified purees were documented to have lower PAS scores for vegetables and meat, but not rice; a significant difference was only found for meat (*p* = 0.024) with a small effect size. Findings were mixed for residue and number of swallows per bolus, with no consistent patterns emerging. In general, this cohort mostly displayed PAS scores of 1 to 3. Two participants were rated with a PAS score of 5, and a PAS score of 8 (silent aspiration) was observed in two participants; with no distinct patterns for meal type or component. Inter-rater reliability was high, intra-class correlation coefficient was 0.961 with a confidence interval of 95% (0.945–0.972) (F (128, 128) = 50.477, *p* < 0.001).

## 5. Discussion

This study compared traditional puree meals in RACFs to modified puree meals, and used mealtime difficulties, consumer feedback, and instrumental assessment of swallowing safety to assess the differences between the two different meal types. Findings indicate that modified purees that take on a gel-like texture and are reshaped to resemble source ingredients are equally accepted by RACF residents compared to traditional purees, and do not compromise airway safety during swallowing. These meals can also be made in large batches in RACFs as part of daily meal preparation procedures, but this will require restructuring of meal preparation workflow, and additional resources and staff training. The current study conducted a 3 h training workshop for the kitchen staff at each participating RACF. Trainers were on hand to supervise meal preparations before the modified purees were served, and a speech therapist was also present to audit the meal before serving. While the procedures were feasible, the additional human resources involved would be difficult to maintain in the long term, and were more reflective of the additional resources required at an implementation stage of a new diet.

Provision of TMDs in RACFs is an area of unique need. A higher proportion of older adults who reside in RACFs tend to experience dysphagia compared to their counterparts who live in the community [23], which means that there is likely a higher need for modified meals that match the dysphagia and nutritional needs of these individuals. At the same time, RACFs need to prepare meals in large quantities to be cost-efficient in their provision of food and drinks for their residents. This creates a dilemma in the division of resources, as addressing one issue may take away from the other. Standardising TMDs is one solution to this dilemma. The IDDSI has provided a useful framework for this, however it can be limiting, especially as new commercial products emerge that modify foods in novel chemical manners, as evidenced by the findings of this study. 

The modified puree meals in this study took on a gel-like texture following mixing of traditional purees with the enzyme. Interestingly, the PAS score for meat (protein) was markedly lower than that of vegetables and rice (starch and fibre) (Table 4). There may be two possible explanations to this. One is that the enzyme gellant affects different ingredients differently, and even when the appearance of different components of the meal is identical, their physical properties may be different when swallowed. The second explanation is that the participants became more fatigued as the VFSS session progressed, and with meat being the last food to be trialled, it was most affected by fatigue. The difference in PAS score between modified puree meat and traditional puree meat at the participants’ most fatigued state suggests that the modified purees may be more easily manipulated and swallowed by a fatigued system. The mealtime observations support the second explanation to some degree, with participants displaying better oral retention and less anterior spillage when eating modified puree meals (Table 2). To examine either explanation, further studies would need to be carried out with a larger sample size with randomised bolus trials, and importantly, a detailed breakdown of the rheological and textural prosperities of foods modified using the methods used in this study.

As a whole, the modified puree was difficult to categorise into any of the IDDSI food levels, and is best described as a transitional food; VFSS findings showed that this texture can be safe to consume for older adults who had been prescribed traditional purees (IDDSI level 4). Similar findings from another study that treated rice purees in a similar fashion using a different commercial product [21] also support the use of gellant enzymes in the sphere of texture modification for dysphagia. This emerging evidence points to the need for research to maintain pace with commercial product development and how these products are used in dysphagia-friendly food modification [28]. This is especially necessary as dysphagia awareness increases globally, and systems such as the IDDSI are suggested to modify regional foods and drinks that may not be comparable to how the system was first designed [29]. The addition of rheological and sensory analysis of food will likely provide insights into how and why certain textures are suited to those experiencing dysphagia [30,31,32].

The findings of this study did not show appreciable differences in the subjective satisfaction and oral intake between the two puree meals. Subjective satisfaction ratings were often difficult to conduct due to the participants’ difficulty in understanding the Likert scale rating. This could be attributed to reduced cognitive function associated with the participants’ medical diagnoses. Research with adults with and without dysphagia has found that pureed fruits, vegetables, and meats are rated more appealing than when the purees are presented in a molded form, as carried out in the current study [33,34,35]. However, the current study’s findings mirror another study that recruited acute-ward patients who struggled with providing responses; subjective ratings did not differ but oral intake amount increased [36]. This raises the issue of conscious and unconscious appeal to consumers with different cognitive capacities, which will need to be further investigated. 

In terms of oral intake, we found a 7.6% increase in mean oral intake amount per meal for modified purees compared to traditional purees, which was borderline significant in statistical analysis (*p* = 0.05). An increase in oral intake for each meal can add up over time and may result in higher nutritional intake, and this would be especially beneficial for individuals consuming purees that have higher water content and relatively lower nutritional content. The findings of this study suggest that meals with texture-modifying products are comparable to traditionally modified meals in acceptability for consumers, but may lead to higher oral intake at mealtimes, which may have longer term benefits if their nutritional content is equal or superior to that of traditional purees.

TMD as the sole form of intervention for dysphagia in RACFs has shown promise for improving body mass index [37]. The studies that produced this positive outcome provided TMDs for periods of 3 to 6 months [38,39], which indicates the need for longer-term implementation of TMDs for nutritional and oral intake effects to be observed. This study trialed the modified puree for three meals across a period of 3 weeks, largely due to logistical reasons as observations and meal preparations needed to be coordinated with the RACFs in the study. A number of external variables were likely to have influenced the mealtime behaviours of the participants during each mealtime observation, as dysphagia is not the only challenge associated with mealtime difficulties in RACFs [40]. The participants’ mood, personal preference for the food served on the day, utensils used, and other factors could have contributed to differences in the outcome measures, and long-term consistent provision of the modified puree would be needed to control for these factors.

### Limitations

The current study was a feasibility and acceptability study with a small cohort of participants living in RACFs. The small sample size of 28 participants limited the depth of analysis that could be conducted on the data collected. Individuals, such as the participants in this study, who need to consume purees in the long-term are likely to be physically frail and may have a ceiling to how much their swallowing and oral intake performance can be improved by external factors such as diet texture modification. The small sample size also limited how much subjective feedback could be collected from the participants regarding their satisfaction of the different meal types, largely due to compromised cognitive function and overall frailty impacting comprehension of questions and verbal communication. In addition, instrumental techniques were limited in this study. The VFSS had low frame rates which prevented quantitative analysis via coordinate mapping of swallowing landmarks, which would have yielded more detailed swallowing information. The rheological properties of the modified puree in this study has not been assessed instrumentally, which would aid in the understanding of how it can affect swallowing, and is an area of research need in this field. Lastly, the current study did not measure nutritional values in the meals observed, which would have addressed one of the major factors to be considered for TMDs. 

## 6. Conclusions

Traditional puree meals prepared in RACFs can be modified using commercial products to improve their appeal to consumers. Modified purees that take on a gel-like texture can be categorised as a transitional food based on the IDDSI framework. The gel-like texture is more difficult for older adults to eat independently, but is easier to retain in the oral cavity while eating. Older adults in RACFs may consume more of the meal when it is a modified puree due to its improved visual and textural appeal, and ease of swallowing, but more robust evidence is needed to confirm this in a larger cohort of participants. The modified puree meals can be safely swallowed by older adults who usually consume traditional puree meals, therefore future research can be carried out with this in mind and seek to clarify the consumer’s experience and long-term nutritional value of these meals.

## Figures and Tables

**Figure 1 geriatrics-06-00108-f001:**
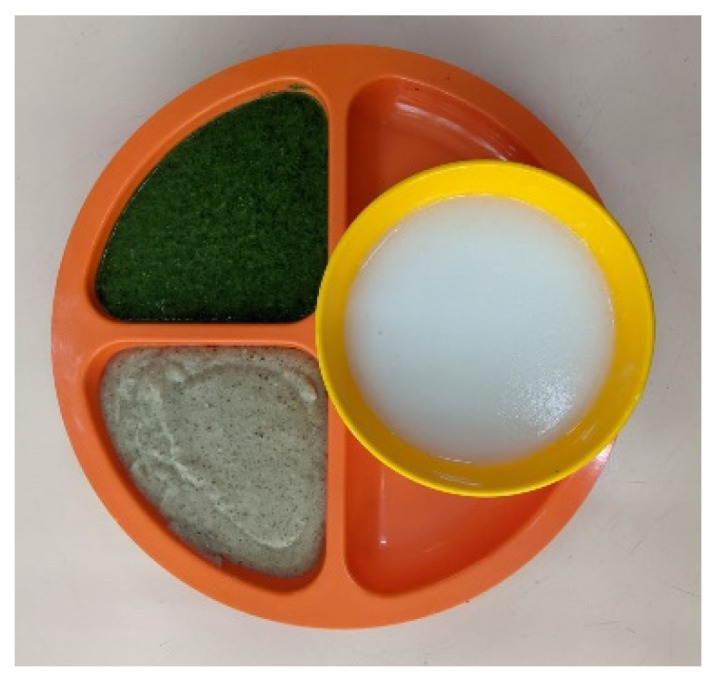
Traditional puree meal served in a residential aged care facility in Hong Kong.

**Figure 2 geriatrics-06-00108-f002:**
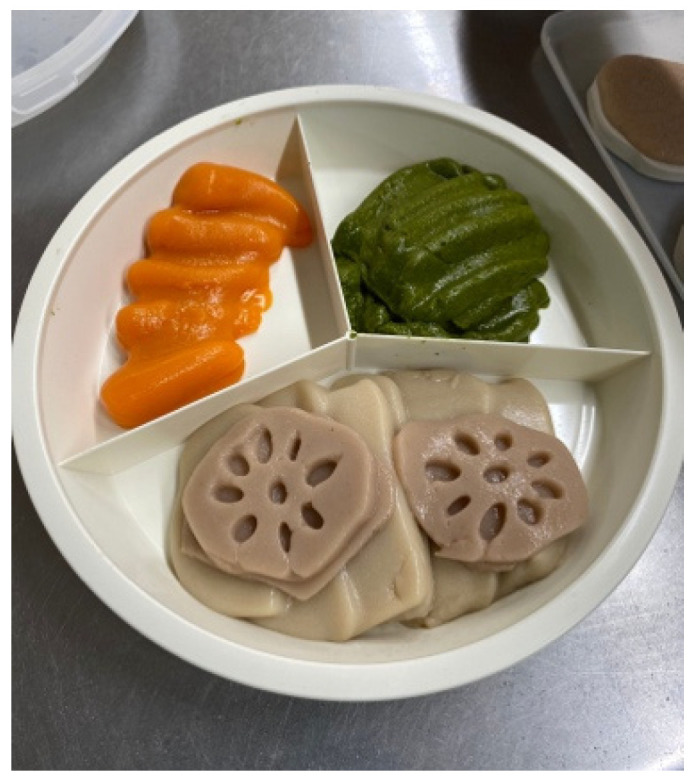
Modified puree meal served in this study.

**Table 1 geriatrics-06-00108-t001:** Demographic and background of the participants of this study.

**Demographics**	Age (Years) ± Standard Deviation, Range	86.1 ± 8.4, 72–104
	Sex	17 females (60.7%) 11 males (39.3%)
**Functional Status**	Level of Care	25 high-level care (89.3%) 3 infirmary level care (10.7%)
	Feeding ability	18 could self-feed (64.3%) 10 required aided feeding during mealtimes (35.7%)
**Diet (Functional Oral Intake Scale—FOIS)**	All puree—FOIS 4	17 (60.7%)
	Puree and soft rice (soft rice equates to IDDSI level 5)—FOIS 5	11 (39.3%)
**Medical Background**	Dementia	13 (46.4%)
	Stroke	12 (42.9%)
	Head and neck cancer	1 (3.6%)
	Parkinson’s Disease	2 (7.1%)
	Respiratory disorder	2 (7.1%)
	Psychiatric disorder	7 (25%)

**Table 2 geriatrics-06-00108-t002:** Mealtime observations for traditional purees and modified purees.

Mealtime Observations	Traditional Puree	Modified Puree	Paired-Samples Test (*p*-Value)	Effect Size
Scooping food up with spoon (0 for no difficulties, 3 for difficulties every meal)	1.07 ± 1.14	1.63 ± 1.21	0.013	−0.33
Bringing food to lips with spoon (0–3)	0.67 ± 0.92	0.59 ± 1.05	0.626	
Oral retention (0–3)	0.70 ± 0.99	0.59 ± 0.93	0.502	
Anterior spillage (0–3)	1.15 ± 1.13	0.52 ± 0.7	0.01	−0.32
Oral residue after meal (0–3)	0.56 ± −0.75	0.41 ±0.64	0.327	
Feeding aid needed (0–3)	0.93 ± 1.24	1.07 ± 1.30	0.212	
Coughs (average per meal)	2.19 ± 3.06, 0–11	2.56 ± 4.14, 0–16	0.479	
Throat clears (average per meal)	1.63 ± 3.36, 0–14	1.44 ± 2.99, 0–11	0.304	
Mean intake (% of entire meal)	72.4 ± 21.84, 23.4–100	79.87 ± 20.34, 20.3–100	0.05	−0.39
Mean meal time (minutes)	26.27 ± 12.17, 10.7–54.7	31.44 ± 17.38, 9–90	0.014	−0.33

**Table 3 geriatrics-06-00108-t003:** Participant feedback for traditional purees and modified purees.

Participant Feedback	Traditional Puree	Modified Puree	Paired-Samples *t*-Test (*p*-Value)
Mean accuracy in identifying meal components (%) (*n* = 25)	24.43 ± 24.68, 0–83.3	25.2 ± 22.94, 0–77.8	0.738
Visual appeal (1 for very unsatisfied, 4 for very satisfied) (*n* = 19)	2.86 ± 0.57, 2–3.7	3.00 ± 0.46, 2–4	0.383
Textural appeal (1–4) (*n* = 14)	3.00 ± 0.34, 2–3.5	3.12 ± 0.25, 2–3.5	0.344
Easy to swallow?	19/21 (90.5%) answered yes	23/23 (100%) answered yes	0.157

**Table 4 geriatrics-06-00108-t004:** Videofluoroscopic swallowing studies (VFSS) for traditional purees and modified purees.

VFSS Outcomes	Traditional Puree	Modified Puree	Paired-Samples Test (*p*-Value)	Effect Size
Penetration Aspiration Scale	Blank/NA			−0.41
Vegetables	2.13 ± 1.77	1.67 ± 1.80	0.244	
Meat	2.36 ± 1.95	1.43 ± 0.65	0.024	
Rice	1.69 ± 1.78	1.75 ± 1.95	0.891	
Vallecular Residue				
Vegetables	1.53 ± 0.99	1.8 ± 1.08	0.271	
Meat	1.93 ± 1.07	2.0 ± 1.24	0.705	
Rice	1.75 ± 1.0	1.69 ± 1.01	0.755	
Pyriform Residue				
Vegetables	0.36 ± 0.48	0.33 ± 0.49	0.655	
Meat	0.71 ± 0.47	0.69 ± 0.85	1.0	
Rice	0.73 ± 0.88	0.53 ± 0.64	0.317	
Mean of Swallows				
Vegetables	1.47 ± 0.67, 1–3	1.67 ± 0.72, 1–3	0.083	
Meat	1.71 ± 0.67, 1–3	1.68 ± 0.64, 1–3	0.739	
Rice	1.59 ± 0.66, 1–3	1.66 ± 0.65, 1–3	0.557	

## Data Availability

Data is available upon request from the corresponding authors.

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
