# Peer review of "Modifying Puree Meals in Residential Aged Care Facilities: A Multi-Centre Feasibility and Acceptability Study"

_geriatrics, 2021, doi:10.3390/geriatrics6040108_

Round 1

Reviewer 1 Report

Thank you for the opportunity to review this paper, which compared the effects of a commercial enzyme gellant on puree food across three residential aged care facilities. There were 28 participants and of these, 16 underwent VFS.

I have a few queries.

The modified puree is considered a feasible approach to improve nutritional intake.

Why was risk of malnutrition or nutritional intake not measured in participants using a standardised tool? It would be interesting to read about the change in participants' nutritional intake after X months for those consuming the modified puree vs the traditional puree. It's difficult to read about compromised nutritional intake in your introduction and discussion when this wasn't explicitly measured in your participants.

Frailty is also referred to in this study but was not measured. Could this measure also be included? 

I think the study objectives (particularly 2 and 3) need revising so that they are more specific. What do you mean by 'effects'? Swallowing safety could be described in more detail in the introduction, including how this relates to pureed food.

It is great to read that a videofluoroscopic swallow study (VFS) was performed in addition to the mealtime observation. Would you consider using quantitative analysis during VFS? Does 10ml reflect a typical bolus size at meal times? Was the 'worst' swallow of the sequence of swallows chosen to measure PAS? (i.e. if there were multiple swallows, which swallow did you choose to rate PAS). Did participants who scored PAS >3 have a known dysphagia or diagnosis known to affect swallowing?

Could you provide some more detail about your statistical analysis? Could you report effect sizes, range, mean/median?

It would have been interesting to compare these results to a 'control group' or a group of RACF without dysphagia, particularly regarding the VFS measures. Would the modified puree diet demonstrate similar results for this 'control group'?

I hope these queries help to support the development of your manuscript.

Best regards

Author Response

Thank you for your time and effort in reviewing our manuscript. Please see below for our responses to each point:

Why was risk of malnutrition or nutritional intake not measured in participants using a standardised tool? It would be interesting to read about the change in participants' nutritional intake after X months for those consuming the modified puree vs the traditional puree. It's difficult to read about compromised nutritional intake in your introduction and discussion when this wasn't explicitly measured in your participants.

We appreciate your focus on nutrition in this topic, and are actually planning to measure nutrition explicitly in our follow up studies now that we have established feasibility and acceptability of the modified purees we trialled in this manuscript.

The modified puree meals were only provided on the 4 occasions of data collection: 3 meals observed and 1 for VFSS. We did not expect any significant changes to nutrition with such limited meals, the focus of the present study was to establish swallowing safety and participant reactions first. Nutrition was raised in the introduction to address all the issues related to texture modified meals, we may not be able to address all the limitations of purees in the present study but we wanted readers to acknowledge their limitations. The discussion section framed nutrition within the limits of our findings, we do not claim that our findings show that any meal improves nutrition, but if there is measured increase in the percentage of a meal that is eaten, it would be reasonable to suggest that over time this may lead to better nutritional intake. We have added the lack of nutritional measurements as an limitation of the study.

Frailty is also referred to in this study but was not measured. Could this measure also be included?

Additional measure of frailty as classified by the Clinical Frailty Scale has been included on page 5 line 213-214. All participants were categorised as severely frail according to the scale.

I think the study objectives (particularly 2 and 3) need revising so that they are more specific. What do you mean by 'effects'? Swallowing safety could be described in more detail in the introduction, including how this relates to pureed food.

Thank you for the suggestion, objectives 2 and 3 have been revised to be more specific.

2. Compare RACF residents’ mealtime experiences of traditional puree and modified puree.

  1. Compare swallowing safety and efficiency be-tween traditional puree and modified puree when consumed by RACF residents.”

Swallowing safety has been expanded on briefly on page 2 lines 45-48:

There are three major negative aspects to the implementation of TMDs. First, although higher viscosity is associated with reduced penetration-aspiration [2], swallowing safety is not guaranteed [3-5] as aspiration of food material into the airway can still occur; and post-swallow residue in the pharynx can also increase with viscosity [2].”

It is great to read that a videofluoroscopic swallow study (VFS) was performed in addition to the mealtime observation. Would you consider using quantitative analysis during VFS? Does 10ml reflect a typical bolus size at meal times? Was the 'worst' swallow of the sequence of swallows chosen to measure PAS? (i.e. if there were multiple swallows, which swallow did you choose to rate PAS). Did participants who scored PAS >3 have a known dysphagia or diagnosis known to affect swallowing?

Unfortunately, the VFSS video frame rate we could access was too low for quantitative analysis of the visuals, this was beyond our control as the external organisation we conducted VFSS with had their own VF protocol that could not be altered. We have added this to limitations on page 11.

We tried bolus sizes of 5mL and 10mL to reflect mealtime bolus sizes, 10mL led to the most salient swallowing performance.

For PAS ratings, the worst score was used. This is specified one page 5 line 209.

All participants exhibited signs and symptoms of dysphagia, this has been added to inclusion criteria on page 2. Additional classification using the FOIS has been added to page 6 line 222 and Table 2.

Could you provide some more detail about your statistical analysis? Could you report effect sizes, range, mean/median?

Data analysis and reporting expanded with the requested information, please see Tables 2, 3 and 4 for added standard deviation ranges, and effect sizes. Ranges for scores within ratings scales were not reported as all rating scales we used were relatively short, i.e. from 1 to 3, and performances already covers the whole range.

It would have been interesting to compare these results to a 'control group' or a group of RACF without dysphagia, particularly regarding the VFS measures. Would the modified puree diet demonstrate similar results for this 'control group'?

This is something we plan to include in follow up studies with more homogenous participants that can be compared. We suspect the modified puree diet is beneficial for very specific populations, now that we have established some basic parameters around this diet we will try to identify those most likely to benefit.

Reviewer 2 Report

Thank you for this important study. Few studies are published with foods as may topic. However, these studies require to be well designed and analysed to help in building the knowledge. This study seems to be a first draft of an article that can be much more.

Your title does not reflect the content. Where is your discussion on feasibility? Did you really assess acceptability?

Please add more details on VFSS used.

Add nutritional values (At least: Energy, Pro, CHO and Fat) of the food served and use these values in discussion. % meal intake is definitely not enough to document nutritional intake. As you mentioned, the litterature points out the dilution of purees.

Include a nutritionnist in your team for analysis.

Please expand on discussion and analysis.

I encourage you to get English language and style support.

Ex: The International Dysphagia Diet Standardization In
tiative (IDDSI) addresses describes the different categorical groups for that foods and liquids can be grouped into for included in dysphagia management.

Line

Comment

2

Residential Aged Care Facilities : could be offensive to some readers. Perhaps use long-term care facilities.

65

Feasibility: Not discussed; please add details pertaining to this objective and how it was assess

or remove as objective and in conclusion.

Throughout the text

Please change speech therapist for dysphagia management specialist as the title of the specialist does not seem associated to speech care or rehabilitation.

It is also more inclusive as international teams vary in composition and expertise.

71

VFSS is an important clinical assessment in dysphagia literature. Few studies report on VFSS and pureed foods.

Please describe how the food/bolus were modified to be radiopaque (barium: liquid, powder, cream; in food? On top?)

Discuss how barium changed texture and these changed were monitored. Discuss how VFSS was associated to results.

79

Ager = aged

81-82

Please rephrase: have seen an increase in interest and capability to use dysphagia products,

but… = have stimulated interest in dysphagia products and increased their use, but …

Nutritional values of the foods must be provided for both Traditional pureed foods and Modified texture foods

Adapt discussion and conclusion with these new data included.

165

Nb of participants must be repeated (n=16; x%) and explain only them?

215-227

Only discuss statistically significant results.

NOTE: Modified pureed foods were statistically more difficult to scoop up and required more

time to eat.

225

More of modified purees were estimated to be eaten by the end of the meal (p=.05), but each meal also took longer for participants to finish (p=.014).

Please recalculate with time included in the calculation (Eg: Mean intake (%) per minute) to

consider the time factor.

228

Oral retention: p=0,10… is this a typo (see Line 222)? If not, correct text at lines

229

Discussion regarding Feedback: No data is statistically significant.

Therefore, rephrase the whole paragraph.

Add to the discussion section : hypothesis of why?

Hint: dementia and stroke… Was this the correct proxy for an assessment of appreciation?

272

Valuable framework? Published yes… but valuable, not proven yet.

291

Add more information regarding IDDSI to the discussion. It is too general at the moment: How did you use IDDSI? What categorization IDDSI tests did you do on both purees? Did you repeat every week? How comparable were your assessment to the label that was initially

provided for your diets?

305-306

Rephrase as this is not an objective statement: Given the abnormal distribution of the data and the small sample size of 28, it is possible that these differences may exist but were not

captured.

309-311

An increase in oral intake for each meal can add up over time and result in higher nutritional intake, and this would be especially beneficial for individuals consuming purees that have higher water content and relatively lower nutritional content.

Where is your data on nutritional values?

Author Response

Thank you for your time and effort in reviewing our manuscript. Please see below for our responses to each point, some comments and questions addressing the same points or sections have been grouped together:

Your title does not reflect the content. Where is your discussion on feasibility? Did you really assess acceptability?

Line 65: Feasibility: Not discussed; please add details pertaining to this objective and how it was assess or remove as objective and in conclusion.

Feasibility was measured by whether each facility kitchen could produce the modified purees following initial training. Additional sentence added to clarify this on page 3 lines 119-120:

“All meals prepared for this study were required to be inspected and approved by a speech therapist who participated in the first two steps before they were served for data collection. This was intended to assess feasibility of the RACF kitchen to produce modified purees following training.

Feasibility result added to page 6 lines 229-232: “ All modified puree meals were inspected an approved by a speech therapist be-fore serving. No meals required further adjusting.”

Discussion of this has been expanded from page 9 line 292 onwards.

“The current study conducted a 3-hour training workshop for the kitchen staff at each participating RACF, trainer were on hand to supervise meal preparations before the modified purees were served, and a speech therapist was also present to audit the meal before serving. While the procedures were feasible, the additional human resources involved would be difficult to maintain in the long term, and were more reflective of the additional resources required at an implementation stage of a new diet.”

Acceptability was measured by the questionnaires administered to the participants after each meal observation. Additional sentences added to clarify this on page 4 line 145. The intended consumers of the modified puree meals are older adults who consume purees, therefore their satisfaction with these meals are an indication of acceptability. To our understanding, acceptability as measured by satisfaction with care/intervention is commonly reported.

Add nutritional values (At least: Energy, Pro, CHO and Fat) of the food served and use these values in discussion. % meal intake is definitely not enough to document nutritional intake. As you mentioned, the litterature points out the dilution of purees.

Include a nutritionnist in your team for analysis.

Nutritional values of the foods must be provided for both Traditional pureed foods and Modified texture foods

Adapt discussion and conclusion with these new data included.

We did not conduct analyses on the nutritional value of the meals. Nutrition was raised as one of the variables that influence the decision to uptake TMDs, as well as one of their limitations, but not the only one. We raised this point to set the background of the study, but we did not set out to measure its nutritional values. We acknowledge that a nutritional analysis would be a valuable aspect for this study, but this was not part of our aims. We aimed to assess the feasibility and acceptability of a modified puree in residential aged care facilities.

I encourage you to get English language and style support.

Thank you for the suggestion, the manuscript has been carefully edited.

Line 2: Residential Aged Care Facilities : could be offensive to some readers. Perhaps use long-term care facilities.

We understand that the term “aged” is part of the discussion about how to better frame the process of ageing and combating ageism, and very much appreciate you raising this in research relating to older adults. “Residential aged care facility” is used in Australia, where the first author drafted this manuscript is based. At the time of writing this response, the Australian Association of Gerontology (an organisation active in the movement against ageism) continues to use this term. The authors take intended and unintended ageism seriously, as researchers and as health care workers who interact with people of all ages and abilities. Please helps us understand your concern around this term.

Throughout the text: Please change speech therapist for dysphagia management specialist as the title of the specialist does not seem associated to speech care or rehabilitation. It is also more inclusive as international teams vary in composition and expertise.

The term “speech therapist” was used as a factual description of the profession of the personnel involved in the study, there are no connotations about specialisation except when stated explicitly in line 112. While we appreciate that different health care professions can be involved in a dysphagia management team, we did not have the privilege to have a whole team, speech therapists were the only health professionals involved in the study.

Line 71: VFSS is an important clinical assessment in dysphagia literature. Few studies report on VFSS and pureed foods.

Please describe how the food/bolus were modified to be radiopaque (barium: liquid, powder, cream; in food? On top?)

Discuss how barium changed texture and these changed were monitored. Discuss how VFSS was associated to results.

Additional section added to address these points on page 5 lines 176-181:

The barium consisted of 340g E-Z-HD (98% w/w) barium and 65mL of water. Purée trials were prepared by combining liquid barium with purée. Nestle ThickenUp starch-based thickener was then added to obtain an IDDSI level 4 texture and was checked by the speech therapist with a spoon-tilt test. To prepare the modified purée trials, each spoonful was mixed with liquid barium. This was then cut into 4 smaller pieces to ensure the bolus was sufficiently coated.”

Discussions of VFSS and meal time observation results discussed on pages 9 and 10.

Line 79: Ager = aged

This typo has been revised, thank you.

line 81-82: Please rephrase: have seen an increase in interest and capability to use dysphagia products, but… = have stimulated interest in dysphagia products and increased their use, but …

Thank you for the suggested edit, this has been revised.

Line 165: Nb of participants must be repeated (n=16; x%) and explain only them?

VFSS was planned for all participants, therefore the methodology for part 2 only reported the procedures. Due to COVID19 outbreak, VFSS sessions were cancelled for many participants as attending hospital for the assessment posed infection risks. Page 8 line 267 onwards addresses this. A percentage of the total cohort has been added in line 266.

Line 215-227: Only discuss statistically significant results.

NOTE: Modified pureed foods were statistically more difficult to scoop up and required more time to eat.

This section has been revised to focus on statistically significant findings.

Line 225: More of modified purees were estimated to be eaten by the end of the meal (p=.05), but each meal also took longer for participants to finish (p=.014).

Please recalculate with time included in the calculation (Eg: Mean intake (%) per minute) to consider the time factor.

Thank you for this suggestion. Our observations showed that participants (and RACF residents overall) did not consume meals with consistent pacing, therefore we elected to consider and calculate the meal as a whole, including time taken to complete the meal.

Line 228: Oral retention: p=0,10… is this a typo (see Line 222)? If not, correct text at lines

This was a typo for anterior spillage, thank you for pointing this out, the value in the table has been corrected to p=.01

Line 229: Discussion regarding Feedback: No data is statistically significant. Therefore, rephrase the whole paragraph. Add to the discussion section : hypothesis of why?

Hint: dementia and stroke… Was this the correct proxy for an assessment of appreciation?

This section has been restructured. Discussion points have been added to page 10 line 320 onwards.

Line 272: Valuable framework? Published yes… but valuable, not proven yet.

“Valuable” has been edited to “useful”. The framework is by no means perfect, but it filled a vacuum and continues to evolve, we hope to see it grow as innovation of dysphagia products grows.

Line 291: Add more information regarding IDDSI to the discussion. It is too general at the moment: How did you use IDDSI? What categorization IDDSI tests did you do on both purees? Did you repeat every week? How comparable were your assessment to the label that was initially provided for your diets?

The IDDSI is designed to be easily used by everyone, therefore its descriptive and measurable criteria to classify textures were used to classify all meals in this study. Auditing of the meals were conducted each time, these details have been expanded on in the method and results sections, as detailed in our responses to your questions regarding feasibility.

Line 305-306: Rephrase as this is not an objective statement: Given the abnormal distribution of the data and the small sample size of 28, it is possible that these differences may exist but were not captured.

This sentence has been deleted.

Line 309-311: An increase in oral intake for each meal can add up over time and result in higher nutritional intake, and this would be especially beneficial for individuals consuming purees that have higher water content and relatively lower nutritional content.

Where is your data on nutritional values?

The discussion section framed nutrition within the limits of our findings, we do not claim that our findings show that any meal improves nutrition, but if there is measured increase in the percentage of a meal that is eaten, it would be reasonable to suggest that over time this may lead to better nutrition al intake, which is what we discussed one page 10, line 354 onwards. We have also added the lack of nutritional measurements as a limitation of the study.

Round 2

Reviewer 1 Report

I think the manuscript has been greatly improved and really appreciate your considerations of the reviewers’ comments.

This study would be of great interest to the local population, as well as other clinicians and researchers working with TMDs, especially within RACFs.

I have a few queries and suggestions for grammatical revisions which are addressed by line numbers.

Queries

33

I wonder if this sentence carrying onto line 34 could be reworded, so it doesn’t read as if older adults are prone to needing TMD (it’s not clear the dysphagia is due to an associated medical cause). Medically unstable patients may be advised to be kept nil by mouth depending on protocols rather than receive TMDs. Could you keep this brief: mealtime difficulties, perhaps due to dentition issues or swallowing difficulties (dysphagia)?

40

Could you briefly define swallowing safety here?

69

There could be a brief sentence introducing the objectives

94 and 97

I think inclusion and exclusion criteria should be written in full sentences

112

The three procedures are explained using different grammar constructions (1 is a title with active voice, 2 is title and passive voice, 3 is passive with no title). This could be revised for consistency.

153

Are the questions in the brief questionnaire available as an appendix?

355

I wonder if this could be simplified as ‘associated with participants’ medical diagnoses’, rather than listing two as these did not affect all of the participants.

362

The details of further investigation (e.g. sample size and controlled / randomisation) have already been mentioned - does this need repeating here? Can you just end with ‘which need further investigation’?

415

In the conclusion ‘harder to eat independently’ - are you referring to the need for feeding support? You may like to change your word ‘hard’ as this is also relevant for texture. Challenging or difficult? I think this bit needs slight alteration so that it does not contradict with line 418 ‘ease of swallowing’.

Suggestions regarding grammar/spelling/typos

25: nutritional intake

26: swallowing safety

63: has shown

66: consumers’

239: inspected and approved

257: was also less unintentional

263: statistically

302: RACF. Full stop - new sentence - A trainer was or Trainers were

317: remove and? Don’t think and is needed

341: This emerging

357: rated more appealing

358: replace like with ‘form, as observed in the current study’ (or words to that effect)

360: replace - with ; subjective ratings

404: reword fined grained (?) swallowing information

Author Response

Thank you for the detailed feedback for the manuscript's language, this is extremely appreciated. We have made edits according to your suggestions, as shown by the track changes in the attached file.

Regarding the query about including the brief questionnaire as an appendix, we have elected not do so as further details of the questionnaire are described from lines 165 to 176. However, we have edited line 165 to be more explicit that the subsequent descriptions are for the questionnaire already mentioned.

Reviewer 2 Report

Thank you for resubmitting promptly and addressing the comments with rigour.

It would be required to document how the foods were assessed for feasibility in the kitchen and categorized, if done (IDDSI assessment at the moment?) if you wish to maintain any reference to IDDSI in the discussion.

Please see comments on the revised document.

Author Response

Thank you for your ongoing effort in reviewing our manuscript. Please see below and the track changed document for our address of specific comments. 

Suggestions to use initials instead of "speech therapist"

Edits have been made throughout the paper to amend this where possible. One speech therapist who assisted in the research is not in the author list, therefore initials could not be used. We also felt it was integral to the methodology to acknowledge the qualification of the individual making the judgements, at least at first mention before falling back to initials. 

Comment about randomisation of bolus trials for VFSS

Yes, this is a short coming in the study, thank you for pointing this out. We have addressed this at the beginning of page 10 in discussions.

Inter-rater reliability

Inter-rater reliability was calculated by having a second rater independently rate 10% of the VFSS footage. This has been described on page 5, section 3.2.2.

Comment in discussion to include feasibility testing in section 2.2

These details are described in section 2.2., specifically as a 3 step processes. 

Comment in discussion: The IDDSI framework was used in the context of the bolus preparation for the VFSS but could not be correlated to foods formulated in the kitchen. Furthermore, it can be limiting,

The IDDSI framework was used as a framework to classify all meal textures in this study. Traditional purees could be categorised easily, but the modified puree could not be. The framework can be used to categorise texture-modified foods in general, which is why we write that it is "a useful framework" (page 9), but new and emerging products such as the one used in this study is creating textures not easily categorised. This is why our discussion states that the framework has limitations. 

We assessed the modified puree against the IDDSI framework and found it did not belong to any of the solid food categories. It's best classified as a transitional food. We have added an additional paragraph in methods to describe this, and amended relevant statements in the discussion (pg 10, line 321) and conclusion (pg 11, line 390) accordingly.

New paragraph in methods: page 3 line 131-133

"The modified puree was categorised as transitional foods using the IDDSI framework. It had a firm texture but would transition into a smooth pureed form after temperature and minimal pressure were applied."

Additional change to figure 1

We changed the photo for figure 1 to a new photo of traditional purees. (bottom of page 3) The new photo presents traditional purees in the same plate as the modified purees in figure 2, and does not contain the different bowls and cups of water and soup that were not assessed in this study.